# Periodic Signal Recovery with Regularized Sine Neural Networks

**David A. R. Robin**                                    DAVID.A.R.ROBIN@GMAIL.COM
**Kevin Scaman**                                          KEVIN.SCAMAN@INRIA.FR
**Marc Lelarge**                                          MARC.LELARGE@ENS.FR
*INRIA - École Normale Supérieure - PSL Research University*

**Editors:** Sophia Sanborn, Christian Shewmake, Simone Azeglio, Arianna Di Bernardo, Nina Miolane

## Abstract

We consider the problem of learning a periodic one-dimensional signal with neural networks, and designing models that are able to extrapolate the signal well beyond the training window. First, we show that multi-layer perceptrons with ReLU activations are provably unable to perform this extrapolation task, and lead to poor performance in practice even close to the training window. Then, we propose a modified training procedure for two-layer architectures with sine activations with a more diverse feature initialization and well-chosen non-convex regularization, that is able to extrapolate the signal with low error well beyond the training window. This procedure yields results several orders of magnitude better than its competitors for distant extrapolation (beyond 100 periods of the signal), while being able to accurately recover the frequency spectrum of the signal in a multi-tone setting.
**Keywords:** Machine Learning, Periodic Signals, Frequency Estimation.

## 1. Introduction

Real-world signals, such as weather forecasts, biological data or financial indicators, often exhibit periodic patterns. These patterns are notoriously difficult to capture with standard neural networks such as ReLU networks (Parascandolo et al., 2017; Eger et al., 2019), thus making deep learning approaches rather unsuccessful for these prediction tasks. While most of the literature focused on using activation functions that exhibit a periodic behavior (Parascandolo et al., 2017; Eger et al., 2019; Ziyin et al., 2020; Mehta et al., 2021), our analysis shows that this approach is, in essence, *necessary yet insufficient* for the purpose of distant extrapolation. More precisely, we show that ReLU networks are fundamentally unable to learn periodic signals, and changing the activations to sine functions requires additional work to ensure proper training. In this work, we propose a novel regularization for sine neural networks that is able to accurately learn the frequency spectrum of a target periodic signal, and thus extrapolate it well beyond the training window.

**Sine-based architectures.** Several architectures using sinusoids as activation have already been proposed in various contexts. For the purpose of image representation, Tancik et al. (2020) introduced a sine-based mapping with static weights used as a preprocessing step to circumvent the issues of ReLU-based networks in learning high frequency components. In a relatively similar setting, but with learned frequency weights, Sitzmann et al. (2020) introduced a sine-based multi-layer architecture to encode various types of signals with high-frequency components. Such changes in architecture are shown to empirically overcome

the problem of learning high frequency components, typically encountered using standard ReLU-based networks (see for instance Jacot et al. (2018) for estimations of the speed at which each component is learned, or Rahaman et al. (2019) for a more targeted discussion of the spectral bias towards lower frequencies). Contrary to the piecewise-linear functions learned with ReLU activations, such choices have the representation power sufficient to learn periodicity present in the data. To investigate this possibility, we construct a one-dimensional well-defined problem where the underlying symmetry must be discovered during training.

**Problem setup.** We consider the problem of recovering a periodic function given observations only on a bounded interval, where the period is unknown. Under the condition that at least two periods have been observed, this problem is always well-defined, however the recovery of the periodic function on the full real line requires one to uncover the underlying symmetry, which can be significantly harder than fitting the data on the observed interval.

For all the following, let $T \in \mathbb{R}_+^*$ be a fixed but unknown period, and $R \in \mathbb{R}_+^*$ be a fixed window size. For all $t \in \mathbb{R}_+^*$, let $\mathcal{F}_t \subseteq (\mathbb{R} \to \mathbb{R})$ be the set of continuous $t$-periodic functions, i.e. satisfying $\forall x \in \mathbb{R}, f(x + t) = f(x)$. Note that these sets of functions are not disjoint. We will assume in all the following that at least two periods have been observed, that is to say $R \geq 2T$. Without this condition, recovering the entire function is clearly impossible, and this condition alone is sufficient to make the problem well defined: If $f^* \in \mathcal{F}_T$ with $T \leq R/2$, then the squared-error minimizer is unique (see Appendix A for a proof)

$$\underset{f \in \cup_{t \leq R/2} \mathcal{F}_t}{\operatorname{argmin}} \mathbb{E}_{x \sim \mathcal{U}(0,R)} \left[ (f(x) - f^*(x))^2 \right] = \{f^*\} \tag{1}$$

The difficulty of this problem does not come from a plethora of indistinguishable minima, as is customary in machine learning, but from the unusual structure of the assumption. Although each $\mathcal{F}_t$ is a vector space, their union is not, for the sum of two periodic functions is not periodic in general. It also lacks an easily exploitable convexity to treat it as an optimization problem that we could easily solve. Moreover, ignoring the periodicity hypothesis and just fitting a function $g : [0, R] \to \mathbb{R}$ to the segment, with a neural network for instance, renders the problem ill-defined, for the loss is supported only on the segment and does not specify how to extend the learned function to the entire real line.

## 2. Proposed architecture and training procedure

### 2.1. Regular Sine-based neural networks

We experiment with two-layer networks with sine as activation ($\sin : \mathbb{R} \to [-1, +1]$). For a width $m \in \mathbb{N}^*$, we initialize independently at random $\omega_i \sim \mathcal{N}(0, 1)$, $a_i \sim \mathcal{N}(0, 1/\sqrt{m})$ and $b_i \sim \mathcal{N}(0, 1/\sqrt{m})$, to get weights $(\omega, a, b) \in \mathbb{R}^m \times \mathbb{R}^m \times \mathbb{R}^m$. This ensures $\sum_i \mathbb{E}[a_i^2] = 1 = \sum_i \mathbb{E}[b_i^2]$, so the initial response is bounded. We train $(\omega, a, b)$ by gradient descent to minimize the empirical loss, using as prediction function $F_{\omega,a,b} : x \mapsto \sum_{i \in [m]} a_i \sin(\omega_i x) + b_i \cos(\omega_i x)$.

### 2.2. Modified initialization and regularization

This minimal modification of the traditional multi-layer perceptron has had little success, because it does not seem to converge to approximations as good as the ReLU-based neural networks. We show that by carefully altering the initialization procedure and regularization,

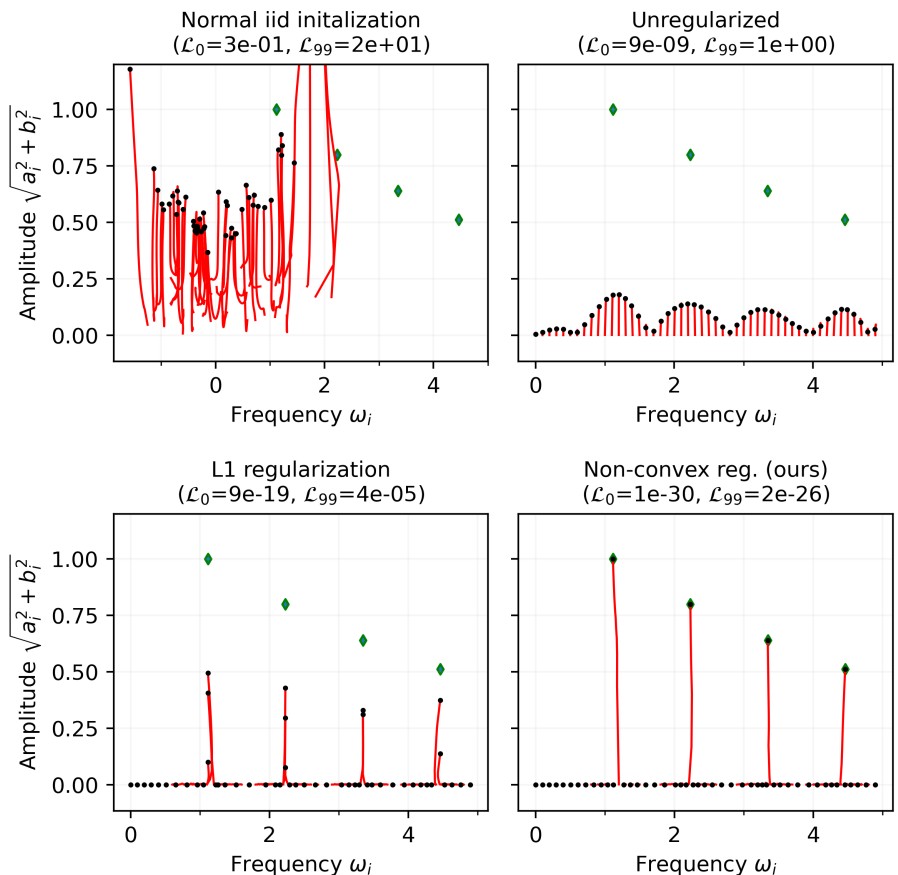

Figure 1: Neuron trajectories in the Fourier half-plane. Black dots indicate the final position of each neuron, red trails depict the trajectory of each neuron during training. Green diamonds indicate an optimal configuration representing the signal exactly (though this parametric global minimum is not unique)

this architecture is capable of distant extrapolation of periodic functions, beating the ReLU-based networks by several orders of magnitude.

To ensure diversity of the initial frequency weights, we initialize them at regular intervals $\omega_i = i\,\delta$, for $i \in [m]$ ($\delta = \pi/R$ in experiments, and there is no need for negative frequencies by symmetry). We initialize the amplitude weights at zero $(a, b) = (0, 0)$. To promote sparsity in the amplitudes, and avoid interferences between very close but distinct frequencies, we use a non-convex sparsity-promoting penalty (with $\varepsilon = 1e\text{-}20$ a small constant for differentiability)

$$\mathcal{R} : (a, b) \mapsto -\sum_{i \in [m]} \exp\left(-\sqrt{a_i^2 + b_i^2 + \varepsilon}\right) \qquad (2)$$

Note that we are not interested in finding the minimum to a different (regularized) loss function, we are merely trying to steer the trajectory of weights toward relatively

sparser amplitudes *during* training. For this reason, we instead use the time-dependent loss $(t, w, a, b) \mapsto \mathcal{L}_0(w, a, b) + e^{-t/\tau} \mathcal{R}(a, b)$ (with $\tau = 10$ in experiments), that is a weighted sum of the empirical loss $\mathcal{L}_0$ and our regularizer $\mathcal{R}$, but whose weight tends to zero during training, such that we truly converge to a minimum of the empirical loss alone when possible.

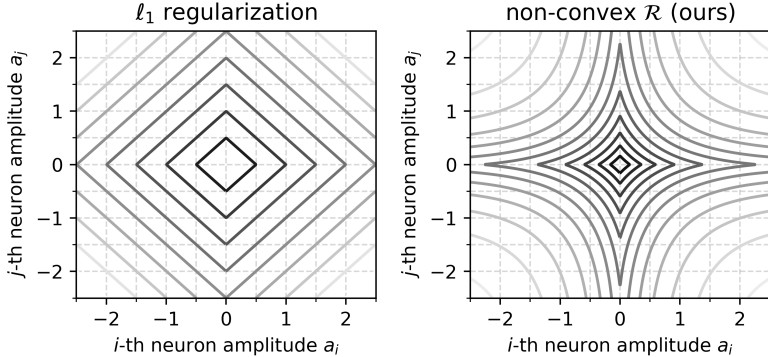

Figure 2: Level sets of regularizations on the $(a_i, a_j)$ slice (with $(b_i, b_j) = (0, 0)$ for simplicity). The sparsity-inducing effect, with $\mathcal{R}$-suboptimality of $a = (1, 1)$ when $\omega_i = \omega_j$ is visible on the picture, $a = (2, 0)$ and $a = (0, 2)$ having lower regularizer values.

In the following sections, we explain the reasons that motivated these choices, starting with the need for a different periodicity-compatible activation function, and then how to fix the training problems that this change induces.

### 2.3. Limits of universal approximation on compacts, the need for sines

A historically strong argument in favor of the representation of functions as neural networks is the theorem of universal approximation on compacts (Cybenko, 1989; Barron, 1993; Leshno et al., 1993), stating that for any continuous function with a compact domain, there exists functions arbitrarily close to it that are representable as a neural network, provided only that the activation function is non-polynomial. This property is not sufficient for periodic recovery, as the followng proposition shows.

**Proposition 1** *If a function $f : \mathbb{R} \to \mathbb{R}$ is representable as a multi-layer perceptron with ReLU activations and finitely many weights, and if $f$ is periodic, then $f$ is constant.*

**Proof** Such a function is piecewise linear with finitely many pieces (see e.g. Hanin and Rolnick, 2019). Thus there exists a constant $B > 0$ such that it is affine on $[B, +\infty[$. If it is both periodic and affine on that interval, then it must be constant on that interval, thus it is constant on $\mathbb{R}$ by periodicity. ∎

This proposition does not constitute a problem in itself, because it remains possible that the learned function is not perfectly periodic, yet remains a very good approximation for a large number of periods after the observed interval, which would suffice in all practical

applications. However, our experiments demonstrate that such networks fail to extrapolate to even a single period outside the training interval.

### 2.4. Problems of the regular sine-based architecture

As presented in the previous section, the sine-based architecture is strikingly similar to a two-layer perceptron with only a modification in activation functions, and this is sufficient to fix the representation problem, all periodic functions are approximable on the whole real line by networks of this form. However, experiments show that training this modified architecture as-is still fails to extrapolate outside the training interval. We identify several problems and propose modifications to the training procedure to fix each of these issues.

**Diagnostic of ill-convergence.** Figure 1 depicts the trajectories of neuron weights over time with two different initializations. A weight $(w, a, b) \in \mathbb{R}^{m \times 3}$ is depicted as a set of $m$ points $(w_i, a_i^2 + b_i^2) \in \mathbb{R} \times \mathbb{R}_+$, together with the corresponding red trail for their evolution over time. The green diamonds correspond to an optimal set of weights, exactly representing the signal. As seen in the first case of Figure 1 (top left), independent initialization of frequency weights ($w \in \mathbb{R}^m$) with a normal distribution barely ever produces very high frequencies, which become harder for the network to learn (a long "distance" has to be traveled by the weights), a more diverse initialization appears preferable.

This problem of absence of high frequencies at initialization is common to all network initializing features weights independently with too low a variance, typically reducing the variance with the number of neurons. For instance, to deal with this issue, SIREN architectures using an initialization of $w_i \sim \mathcal{U}(-\sqrt{6/m}, +\sqrt{6/m})$ with $m \in \mathbb{N}^*$ the number of neurons, have to resort to a hardcoded constant $\omega_0 = 30$ (Sitzmann et al., 2020, section 3.2) in the activation $x \mapsto \sin(\omega_0 x)$, tuned to the considered applications and chosen network sizes, to ensure that the higher frequencies are present at initalization.

**Additional regularization to avoid interferences.** The trajectories of gradient descent have several times been linked with a form of "$\ell_2$-inertia", see for instance Gunasekar et al. (2018). Indeed in Figure 1, we see that many neurons have a small but non-zero amplitude. While this may not be a problem in the training interval, it tends to cause unwanted wiggling outside the training interval, because these frequencies are usually not rationally linked and thus do not produce exactly periodic functions but only functions that look approximately periodic on the training interval, see for instance learned signals in Figure 3. This issue can be fixed by promoting sparsity in the amplitudes, but some care must be taken to not introduce a phase bias by the choice of regularization. Indeed, a change of phase in the signal corresponds to a rotation in the $(a, b)$ plane. Therefore, to avoid the phase-bias, the regularization chosen must depend on $r_i^2 = a_i^2 + b_i^2$ but not $a_i$ and $b_i$ individually. As shown in Fig. 1, an $\ell_1$ regularization $\sum_i r_i$ does induce sparsity, but does not distinguish between two very similar neurons, which causes issues on large extrapolation intervals, which are of particular interest in these experiments (see loss values in Fig. 1). To avoid these interferences, we use the non-convex penalty $\mathcal{R}(a, b) = -\sum_i \exp(-r_i)$, which even for $\omega_i = \omega_j$, forces either $r_i$ or $r_j$ to be null, where the $\ell_1$ penalty allows any combination with constant sum.

We emphasize that this problem is very different from the previous point. While non-independent initialization is used to ensure *presence* of high frequencies in the learned signal,

this regularization ensures *frequency localization* after training. The effect can be observed in Figure 1. On the top right panel, there is a diverse initialization, thus higher frequency components are present, but the support is spread out, many frequencies have non-zero amplitude. As a result, the training error $\mathcal{L}_0$ is very low, but the extrapolation error $\mathcal{L}_{99}$ remains high, because the learned function is not periodic. See Appendix B.1 for a discussion of why. In contrast, on the bottom panels, the learned signal is much more localized in frequency, corresponding to the desired periodic behavior.

## 3. Experiments

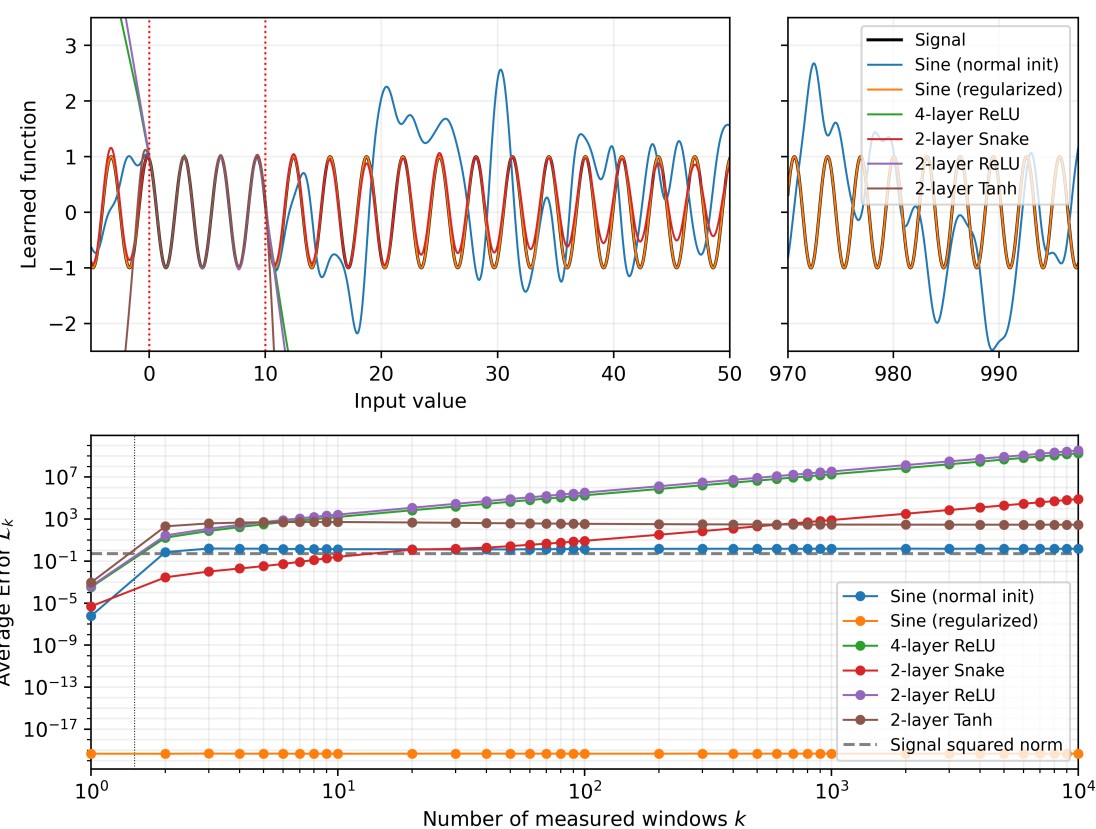

Figure 3: Learned function and corresponding error for each architecture (single tone signal).

**Evaluation methods.** We use $R = 10$ as window size for our experiments. For each experiment, we select a period $T \in [0, R]$, a $T$-periodic function $f^* \in \mathcal{F}_T$, a number of observed samples $n \in \mathbb{N}^*$. We then sample independently uniformly at random on $[0, R]$ a collection of $n$ samples $(x_i)_{i \in [n]} \in [0, R]^n$. Along with these inputs, algorithms are provided with the noiseless response $(y_i = f^*(x_i))_{i \in [n]} \in \mathbb{R}^n$. After training, we evaluate the empirical error

$\mathcal{L}_0 : f \mapsto \frac{1}{n}\sum_i (f(x_i) - y_i)^2$, and the $k$-windows expected error $\mathcal{L}_k : f \mapsto \mathbb{E}_x[(f(x) - f^*(x))^2]$ where $x \sim \mathcal{U}(0, kR)$, for various values of $k \in \mathbb{N}^*$. In similar settings in machine learning, the empirical error $\mathcal{L}_0$ is referred to as "train loss", and the 1-window expected error $\mathcal{L}_1$ as the "test loss", for its underlying distribution corresponds to that of the samples. However, the performance we are truly interested in is the $k$-windows expected error $\mathcal{L}_k$ for large values of $k$ (typically $k \geq 10$ at the very least for simple signals), for which we can really claim that the periodicity assumption has been exploited.

**Baseline architectures evaluated**  We experiment with multi-layer perceptrons with ReLU activations $((\cdot)_+ : x \in \mathbb{R} \mapsto \max(0, x))$. For a number of hidden layers $L \in \mathbb{N}^*$, and a width $m \in \mathbb{N}^*$, we initialize independently at random $\theta_i^0 \sim \mathcal{N}(0, 1)$ for $i \in [m]$ for the first layer, $\theta_{i,j}^k \sim \mathcal{N}(0, 1)$ for $(i, j) \in [m] \times [m]$ for layer $k \in \{1, \cdots, L - 1\}$, and $\theta_i^L \sim \mathcal{N}(0, 1/\sqrt{m})$ for $i \in [m]$ for the last layer. For an input $x \in \mathbb{R}$, we set the initial activation $z^0 = x$, then recursively $z_i^{k+1} = \max(0, \theta_i^k \cdot z^k)$ for $k < L$, and use as prediction function $F_\theta(x) = \sum_i \theta_i^L z_i^L \in \mathbb{R}$. We then train the weights $\theta \in \mathbb{R}^m \times (\mathbb{R}^{m \times m})^{L-1} \times \mathbb{R}^m$ by gradient descent (Adam optimizer, step size 1e-4) to minimize the empirical loss. We also perform experiments with other non-linearities such as tanh, and the snake $x \mapsto x + \frac{1}{a}\sin^2(ax)$ activation (Ziyin et al., 2020) with $a = 27.5$ (authors recommend $a \in [5, 50]$).

**Single-tone periodic recovery**  We set $\omega^* = 2$ and signal $f^* : x \mapsto \cos(\omega^* x + \pi/12)$. We use $n = 1000$ samples. Figure 3 shows the performance of each method, along with the function learned. All methods achieve reasonably low error on the training set, and comparable "test" error $\mathcal{L}_1$ (which is almost identical because a thousand-point average on an interval of length 10 is a very good estimate). However, the ReLU architecture fails to uncover the symmetry and produces linear extrapolations outside the training interval.

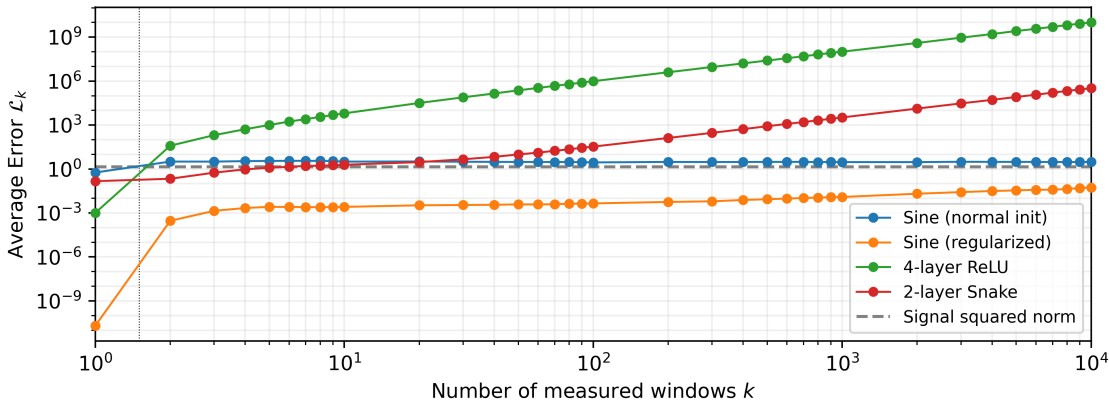

Figure 4: Error of the various architectures with a more complicated signal.

**Multi-tone periodic recovery**  With $n = 1000$ samples as previously, we set $\omega^* = 2$, $K = 25$ and $a_k = (-0.8)^k$ for $k < K$, to construct the more complicated periodic target signal $f^* : x \mapsto \sum_{k<K} a_k \cos((k+1)\omega^* x + \pi/12)$. The results are depicted in Figure 4.

The relative ordering of methods by reconstruction accuracy is essentially unchanged. The more complicated signal induces higher average errors even on the observed interval (the in-interval "test" error $\mathcal{L}_1$ is order of magnitudes larger than for the simple sine wave of Fig. 3), and the errors are even larger when more periods are observed. One interesting particularity in this experiment is that we observe increasing errors even for the regularized sine network (albeit at a different rate). This seems to be caused by the inexactitudes in the learned harmonics. Roughly speaking, we would want $\omega_k = k\,\omega_1$ essentially for a perfectly periodic signal, whereas the learned $\omega_k \approx k\,\omega_1$ yield small deviations from this periodic behavior which are nearly invisible in the first few periods but start to show when more periods are observed.

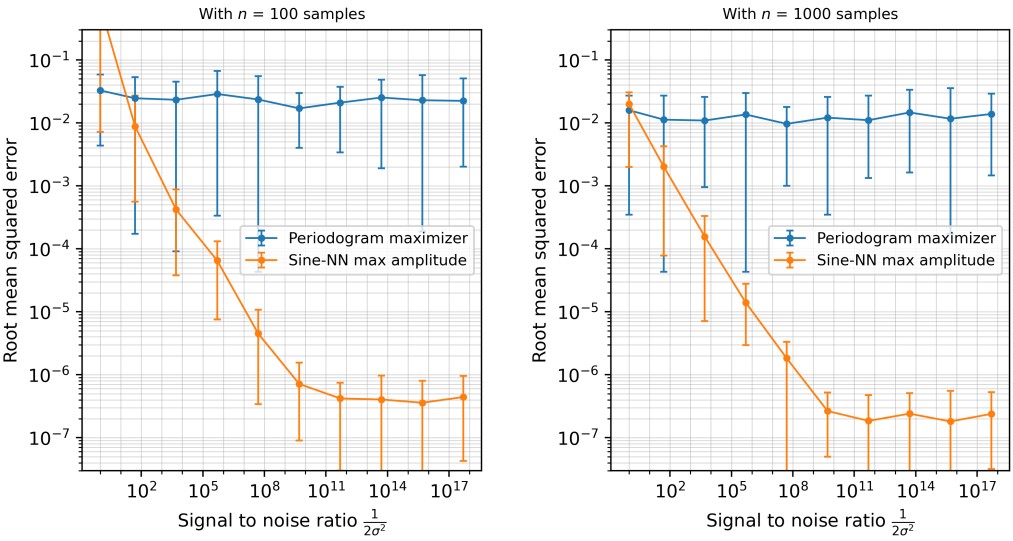

Figure 5: Frequency estimation performance with respect to noise level

**Frequency estimation.** In Figure 5, we compare the frequency estimation capability of our method with periodogram maximization (Kootsookos, 1993) for $n = 100$ and $n = 1000$ samples on a window of size $R = 10$, with independent identically distributed noise (normal distribution with mean zero and standard deviation $\sigma$ for an amplitude of 1.0, yielding a signal-to-noise ratio of $1/2\sigma^2$). For our method, we estimate the whole signal in $10^5$ iterations, then return the frequency $\omega_k$ associated with the maximum final amplitude $k = \text{argmax}_i(a_i^2 + b_i^2)$. Note that the computational cost of the algorithms are hardly comparable, ours requires solving an optimization problem every time, whereas periodogram maximization is nearly-instantaneous here. However this may still be useable in a regime where very few data points are available. We depict the root mean squared error over 15 independent runs with true frequency sampled uniformly in $[\omega_0, 15]$ for each run, where $\omega_0 = R/2\pi \approx 1.59$ is the minimal frequency to ensure that at least one period has been observed. Error bars depict the minimum and maximum value of the root mean squared error. Regarding orders of magnitudes, each frequency estimation with Sine-NN in the context of Figure 5 took around 13 minutes in our experiments for $n = 100$ samples and 20 minutes

with $n = 1000$ samples, compared to under a second with periodogram maximizer. We regard these performance considerations as outside the scope of this paper, proper evaluations in controlled environments should be conducted before any conclusions can be drawn. We presented these experiments as simple counterexamples to conjectures in theoretical machine learning, we claim only that they demonstrate plausible usability for the purpose of theorists, not that this architecture and training procedure are usable as-is in realistic tasks.

## 4. Conclusion

We have shown with simple and well-defined examples that neural networks with sine activations, often disregarded in favor of more complicated solutions resembling ReLU activations, require only minimal modifications to yield trainable architectures with competitive performance on the task of periodic signal recovery. While the direct applicability of this architecture to more realistic signals and partially-periodic datasets remains as future work, these experiments provide good sanity checks and baselines for architectures designed to learn periodic trends in larger settings in the future.

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

## Appendix A. Uniqueness of mean squared error minimizer

Let us prove Equation 1 (reproduced below).

$$\underset{f \in \cup_{t \leq R/2} \mathcal{F}_t}{\operatorname{argmin}} \mathbb{E}_{x \sim \mathcal{U}(0,R)} \left[ (f(x) - f^*(x))^2 \right] = \{f^*\}$$

The proof of unique minimizer is almost immediate on $\mathcal{F}_T$ because it is a vector space. The only difficulty in extending it to $\cup_{t \leq R/2} \mathcal{F}_t$ is to avoid picking $t$, then $f \in \mathcal{F}_t$ a minimizer and try to show $t = T$, which does not hold in general, because the sets $(\mathcal{F}_u)_u$ are not disjoint.

**Proof** Let $R \in \mathbb{R}_+^*$. Let $f^* \in \mathcal{F}_T$ be the target, with period $T \in \mathbb{R}_+^*$ such that $T \leq R/2$ (i.e. at least two periods have been observed). Define $\mathcal{L}$ the mean squared error on $[0, R]$:

$$\mathcal{L} : f \mapsto \mathbb{E}_{x \sim \mathcal{U}(0,R)} \left[ (f(x) - f^*(x))^2 \right]$$

We will show that $f^*$ is the unique minimizer of $\mathcal{L}$ in $\cup_{t \leq R/2} \mathcal{F}_t$. First, note that $\mathcal{L}(f^*) = 0$, and $\mathcal{L}$ has only non-negative values, therefore $f^*$ is a minimizer of $\mathcal{L}$, and $f^* \in \mathcal{F}_T \subseteq \cup_{t \leq R/2} \mathcal{F}_t$ because $T \leq R/2$ by assumption. It remains to show that $f^*$ is the unique such minimizer. For the first part, note that if $f \in \mathcal{F}_T$, and $\mathcal{L}(f) = 0$, then $f = f^*$, because $(f - f^*) \in \mathcal{F}_T$ is

a continuous $T$-periodic function null almost-everywhere in $[0, R]$, therefore null on $[0, R]$ by continuity, thus null on $\mathbb{R}$ by $T$-periodicity since $T \leq R$.

Let us proceed to show that all minimizers are in $\mathcal{F}_T$. Let $f \in \cup_{t \leq R/2} \mathcal{F}_t$ be such that $f \notin \mathcal{F}_T$, and let us show that $\mathcal{L}(f) \neq 0$. The assumption $f \notin \mathcal{F}_T$ implies that there exists $x \in \mathbb{R}$ such that $f(x) \neq f(x + T)$. By continuity, there exists an open interval $I \subseteq \mathbb{R}$ containing $x$ such that $\forall u \in I, f(u) \neq f(u + T)$. However, since we have $f \in \cup_{t \leq R/2} \mathcal{F}_t$, there exists $t \in ]0, R/2]$ such that $f \in \mathcal{F}_t$, therefore we can assume without loss of generality that $x \in [0, R/2]$ (otherwise let $v = x - \lfloor x/t \rfloor t$, satisfying $v \in [0, t[ \subseteq [0, R/2[$, and observe $f(v) = f(x) \neq f(x + T) = f(v + T)$ by $t$-periodicity of $f$). Observe now that $(x + T) \leq R$ because $x \leq R/2$ and $T \leq R/2$. Therefore (up to shrinkage of $I$ to ensure $I + T \subseteq [0, R]$ and $I \cap (I + T) = \emptyset$), there is an open interval $I$ of $]0, R[$ such that $\forall u \in I, f(u + T) \neq f(u)$ and $\forall u \in I, u + T \in [0, R] \setminus I$. To conclude, use the identity $(a - b)^2 + (a - c)^2 \geq \frac{1}{2}(b - c)^2$, which holds for all $(a, b, c) \in \mathbb{R}^3$, where $a = f^*(u) = f^*(u + T)$ by $T$-periodicity of $f^*$, to get

$$\mathcal{L}(f) = \frac{1}{R} \int_{[0,R]} (f^*(u) - f(u))^2 \, \mathrm{d}u$$
$$\geq \frac{1}{R} \int_I \left( (f^*(u) - f(u))^2 + (f^*(u + T) - f(u + T))^2 \right) \mathrm{d}u$$
$$\geq \frac{1}{2R} \int_I (f(u) - f(u + T))^2 \, \mathrm{d}u > 0$$

$\blacksquare$

### A.1. One observed period is not sufficient to learn the period

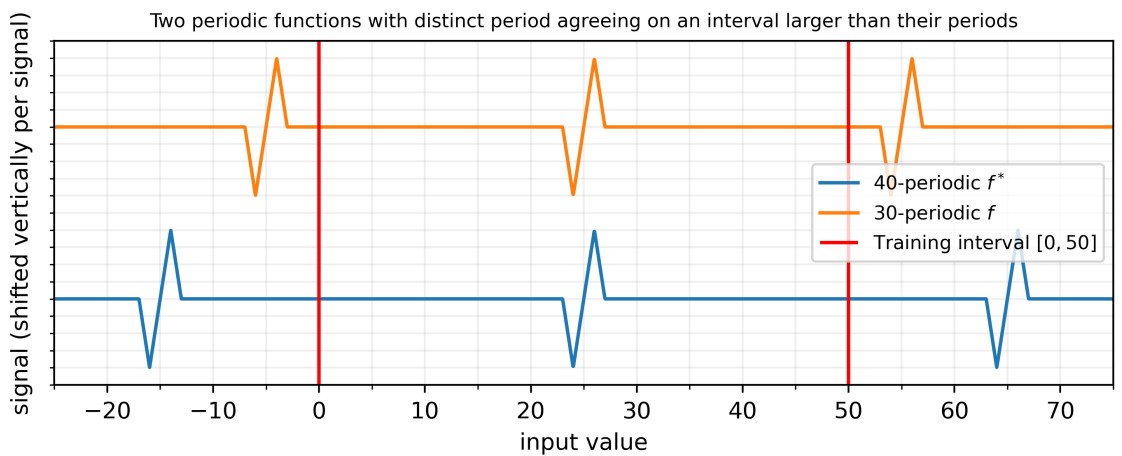

Figure 6: Counterexample to uniqueness if only one period is observed instead of two

## Appendix B. Ablation study

### B.1. Effect of regularization on frequency localization

Learning a two-layer sine network representation for the function $f^* : x \mapsto \sin(\omega^* x + \varphi)$ would be trivial if we had access to values on $\mathbb{R}$, we would just compute its Fourier transform. Since it is composed of a single sine wave in input-space, this would lead to a dirac in frequency-space, located at the sine frequency $\omega^*$. However, we have access only to values on the interval $[0, R]$, thus for all practical purposes, $f^*$ is indistinguishable from the truncated function $(g : x \mapsto \mathbb{1}(x \in [0, R]) \sin(\omega^* x + \varphi))$, identical to $f^*$ on $[0, R]$, but zero outside.

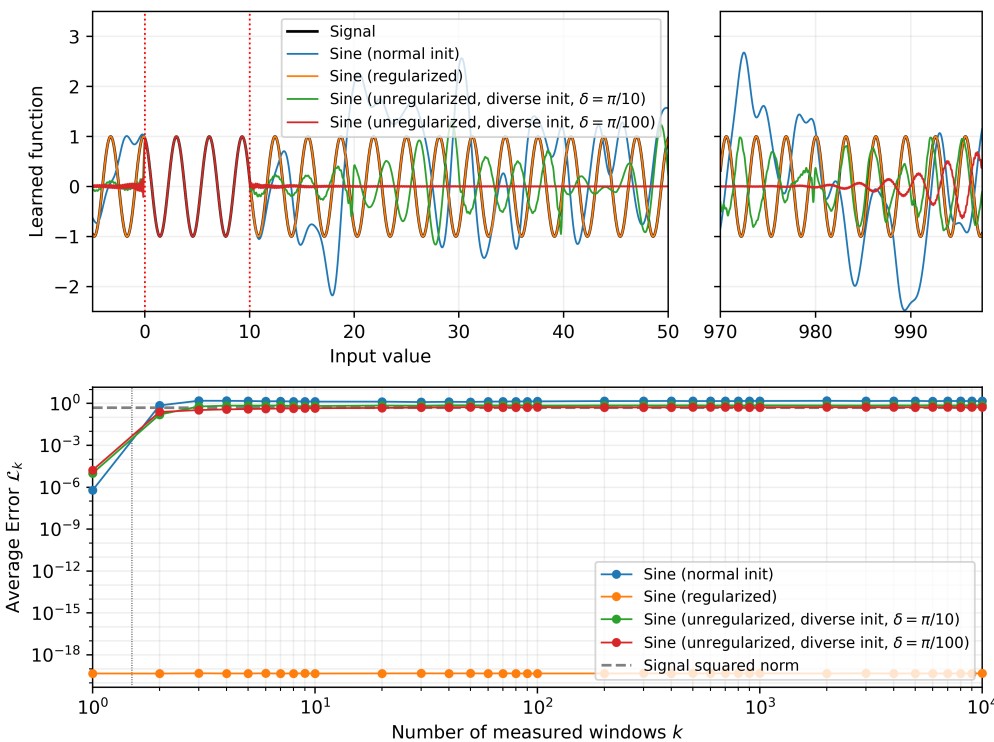

Figure 7: Single-tone experiment with sine network variations

The Fourier transform of $g$ is a convolution of the Fourier transform of $f^*$ with a cardinal sine filter corresponding to the truncation to the interval. Thus, while the Fourier transform of $f^*$ is perfectly localized in frequency because it corresponds to a periodic function in input-space, the Fourier transform of $g$ is "spread out", with many non-zero amplitudes around $\omega^*$ corresponding to the convolution with the filter. Without regularization, the network tends to learn $g$ rather than $f^*$ (both would lead to zero error, but the $\ell_2$ norm of amplitudes for $g$ is smaller than that for $f^*$). This can be observed in Figure 1 (top right) in the Fourier half-plane, and in Figure 7 in input-space: as the gap between frequencies at initialization shrinks, the learned function gets closer to $g$, with values of zero outside the training interval.

