# OpenReview forum: "Periodic Signal Recovery with Regularized Sine Neural Networks"
_NeurIPS.cc/2022/Workshop/NeurReps — NeurReps 2022 Poster_

### Official Review · Reviewer_Jao5 · 2022-10-10
**Concerns about the similar idea of using sine activation and needs more comparisons with other methods**

**Confidence:** 4
**Soundness:** 2
**Presentation:** 3
**Contribution:** 1
**Overall Rating:** 3

**Summary:**

The submission proposes an architecture using sine activation functions along with a non-convex regularization to learn a periodic one-dimensional signal. The proposed method is able to extrapolate the periodic signal with low error well beyond the training window.

**Questions:**

In Proposition 1, the author states that a MLP with ReLU activation functions and finite parameters cannot represent a periodic function other than a constant function. What if you write $sin(x) = \sum_{n=0}^{\infty} (-1)^{n}\frac{x^{2n-1}}{(2n+1)!}$? Actually simple MLP with ReLU activation functions is able to represent a periodic function. The real problem with ReLUs is that it's difficult to learn the high frequency components of the function, see more details in [3].

[3] Rahaman, Nasim, et al. "On the spectral bias of neural networks." International Conference on Machine Learning. PMLR, 2019.

**Limitations:**

The authors compare their methods with very simple MLP models, it needs more study both empirically and theoretically.

Empirically, the authors should compare their method with SIREN [1] and Fourier feature mappings [2].

Theoretically, the authors should also consider relate their method to spectral bias.

**Recommended Decision:**

1: Reject

**Relevance:**

3: Solid fit

**Strengths And Weaknesses:**

Recovering a periodic function from observations only on a bounded interval is an interesting problem. The authors uses sine activation functions with diverse weight initializations to learn periodic functions and add a non-convex regularization during the training. The method is reasonable and straightforward, however, this method is NOT NEW, actually it has been explored in many papers and this paper fails to report, compare and discuss the existing literatures:

[1] Sitzmann, Vincent, et al. "Implicit neural representations with periodic activation functions." Advances in Neural Information Processing Systems 33 (2020): 7462-7473.

[2] Tancik, Matthew, et al. "Fourier features let networks learn high frequency functions in low dimensional domains." Advances in Neural Information Processing Systems 33 (2020): 7537-7547.



**Submission Track:**

Proceedings Paper (9 Page)

---

> ### Author Response · Authors · 2022-11-02
> **Response to Reviewer Jao5**
>
> Thank you for your time reviewing this paper, we have incorporated your suggested references in the camera-ready version, and better highlighted key differences in the training procedures. Regarding your question on the Taylor expansion of the sine function into $f_m : x \mapsto \sum_{i < m} (-1)^i \frac{x^{2i + 1}}{(2i + 1)!}$, if the function is to be approximated on a compact $\mathcal{K}$ only, then indeed for every desired precision $\varepsilon > 0$ there exists a number of terms $m$ such that $\forall x \in \mathcal{K}, \lvert \sin(x) - f_m(x) \rvert < \varepsilon$. However, for periodic functions this restriction to compacts is unnecessary (we have enough information on the training interval to recover it completely), so we wish to lift that assumption and learn the target function on the entire real line (i.e. replace with $\forall x \in \mathbb{R}$, for which this strategy of increasing $m$ no longer works), which is why we need to turn to sine activations. We provide in Figure 3 an example target function without high-frequency components (a sine wave), to illustrate this issue of restriction to compacts and why ReLU-based networks fail to extrapolate past the observed interval.

---

### Official Review · Reviewer_Jrzf · 2022-10-10
**Periodic Signal Recovery with Regularized Sine Neural Networks**

**Confidence:** 4
**Soundness:** 3
**Presentation:** 3
**Contribution:** 3
**Overall Rating:** 7

**Summary:**

The authors develop a network architecture and optimization scheme for periodic signals that can generalize well to test points outside the window over which the training is done.

**Questions:**

1) Figures could benefit from subfigure labels - some of the phenomena mentioned would be clearer with text references to the plots particularly with figure 1.

2) The unregularized network (figure 1) performs somewhat well at least in that it captures the frequency peaks, but i wouldn't have expected this. what accounts for the improvement relative to the IID initialized network?

3) for the complicated signal (figure 3), the average error for all networks seems to compound over the measured windows, whereas this doesn't seem to be the case for the sine/regularized network in figure 2. What accounts for this?



**Limitations:**

Yes, the authors mention the periodogram algorithm achieves faster optimization, but they suggest their method is still tenable in data-limited regimes.

**Recommended Decision:**

3: Accept

**Relevance:**

3: Solid fit

**Strengths And Weaknesses:**

Previous work has focused on the necessity of using periodic activation functions for deep learning tasks that involve learning periodic signals. The work's contribution is original in that it shows a fundamental weakness with networks with both ReLU and sine activations that they can't learn periodic signals without additional constraints on the training.

The work provides high quality experimental results that showcase the improvement in performance when using their architecture and regularization technique relative to several control networks.

The work is clear and selects useful examples. Some more development of the mathematical ideas (e.g., the uniqueness of the mean squared error minimizer solution was not obvious to me) would make the work more clear.

The work is significant/relevant to the community in that it addresses learning a problem with an underlying symmetry.

**Submission Track:**

Proceedings Paper (9 Page)

---

> ### Author Response · Authors · 2022-11-02
> **Response to Reviewer Jrzf**
>
> Thank you for your time reviewing this paper, we have added short discussions of the questions you raised in the camera-ready version. There was indeed a mistake on the minimal number of periods required for uniqueness, thanks for catching that. We have updated the claim and added a proof in appendix (with a counterexample to outline the previous mistake).

---

### Official Review · Reviewer_Tq1q · 2022-10-15
**Periodic Signal Recovery**

**Confidence:** 4
**Soundness:** 3
**Presentation:** 3
**Contribution:** 3
**Overall Rating:** 6

**Summary:**

This paper address the problem of recovering 1D periodic patterns in a well-defined setup where observed data lies in a bounded interval spanning at least one period. The goal is not only to predict well within the observed window, but also extrapolate well beyond that. This works shows that ReLU networks perform poorly on this task and a simple theorem supports the fundamental limitation of such networks. The simple architecture of two-layers with frequencies and amplitudes as parameters and sine as the activation function, fails to extrapolate and through a series of experiments, the potential deficiencies are identified and addressed using simple yet effective modifications. First, a more diverse initialization is used for frequencies to ensure better coverage. Second, a non-convex penalty regularizes the amplitudes to be sparse and avoid interference between neurons with close frequencies. The proposal is evaluated on toy problems of single-tone and multi-tone recovery and frequency estimation. With the tailored initialization and specialized regularization, the sine network significantly outperforms baselines, especially on distant points from the training window.

**Questions:**

* Questions:
    1. It is mentioned that the proposal is more expensive than periodogram maximization for the frequency estimation task. Can you please provide some numerical values to compare the computational costs? It is helpful for broader audience.
    2. Can you please include the curve for the sine network with just the proposed initialization in figure 2? It facilitates inspecting the advantage of the new initialization over the simple sine baseline. Also, in the same figure, top panel, some curves cannot be distinguished from each other.
    3. Is it possible to extend the mathematical discussion around the non-convex penalty? L1 regularizer apparently allows for multiple neurons with same frequency to have non-zero amplitudes. However, only a hand-wavy argument is provided in favor of the non-convex penalty. For instance, in case of two neurons with same frequency, putting all the power on one neuron results in smaller penalty than when it is divided. Namely, for a fixed sum $a+b=r$, the minimum of $-\exp(-a)-\exp(-b)$ is achieved when $a=0, b=r$ or $a=r, b=0$.

* Suggestions:
  1. I highly suggest that you add a more comprehensive review of the previous work.
  2. Please add numbers to the equations


**Limitations:**

As mentioned above, a more rigorous analysis of why the non-convex penalty term helps avoiding interference between neurons with close frequencies, is missing. Also, adopting regularized sine networks for more realistic data is an important future step.

**Recommended Decision:**

3: Accept

**Relevance:**

3: Solid fit

**Strengths And Weaknesses:**

* Strengths
    1. The work motivates the use of sine activation function by showing the fundamental limitation of ReLU networks in approximating periodic function. This is explained well in Proposition 1 and its follow-up discussion.
    2. A set of experiments are conducted with insightful visualizations in figure 1, and they clearly investigate the issues with initialization and optimization for the sine-based architectures.
    3. The problem setting is clear and the challenges of optimization over periodic functions are described well.
    4. The numerical results provide strong evidence that the proposed work is powerful in extrapolation for the toy problems. As neural networks are notoriously poor in extrapolation, this new property might be of interest to the community.
* Weaknesses:
    1. There is a lack of evaluation on real data that might not be perfectly periodic. This raises some doubts about applicability of the proposal to real data.
    2. The discussion over the related-work is limited.
    3. Sparsity of amplitudes looks critical for the toy problems in the paper. However, it might become problematic in cases where the ground-truth data consists of several sine or cosine terms with dense set of frequencies.

**Submission Track:**

Proceedings Paper (9 Page)

---

> ### Author Response · Authors · 2022-11-02
> **Response to Reviewer Tq1q**
>
> Thank you for your time reviewing this paper, we have incorporated your suggestions into the camera-ready version of this submission. The weakness of applicability remains though, we targeted here a more intuitive understanding of why neural networks fail to learn periodicity, with
>  pictures and simple examples. Functions with dense Fourier support are by construction not periodic, so we considered that out of scope; however, we expect indeed that the question of what to do with data in between, that is somewhat periodic but not perfectly, will be of interest
>  to practitioners, but that performance assessment will require much more experimental evidence than we can provide here.

---

### Decision · Program_Chairs · 2022-10-21

Accept (Poster)